# The Role of the Social Determinants of Health on Engagement in Physical Activity or Exercise among Adults Living with HIV: A Scoping Review

**DOI:** 10.3390/ijerph192013528

**Published:** 2022-10-19

**Authors:** Farhana Safa, Natalia McClellan, Sarah Bonato, Sergio Rueda, Kelly K. O’Brien

**Affiliations:** 1Institute for Mental Health Policy Research (IMHPR), Centre for Addiction and Mental Health (CAMH), Toronto, ON M5S 2S1, Canada; 2Campbell Family Mental Health Research Institute, Centre for Addiction and Mental Health (CAMH), Toronto, ON M5S 2S1, Canada; 3AIDS Community Care Montreal, SIDA Bénévoles Montréal, Montreal, QC H2L 2Y4, Canada; 4Library Services, Centre for Addiction and Mental Health (CAMH), Toronto, ON M5S 2S1, Canada; 5Department of Psychiatry, Temerty Faculty of Medicine, University of Toronto, Toronto, ON M5T 1R8, Canada; 6Institute of Health Policy, Management and Evaluation (IHPME), Dalla Lana School of Public Health, University of Toronto, Toronto, ON M5T 3M7, Canada; 7Department of Physical Therapy, Temerty Faculty of Medicine, University of Toronto, Toronto, ON M5G 1V7, Canada; 8Rehabilitation Sciences Institute (RSI), University of Toronto, Toronto, ON M5G 1V7, Canada

**Keywords:** people living with HIV/AIDS, physical activity, exercise, social determinants of health

## Abstract

Physical activity (PA) and exercise are an effective rehabilitation strategy to improve health outcomes among people living with HIV (PLWH). However, engagement in exercise among PLWH can vary. Our aim was to characterize the literature on the role of social determinants of health (SDOH) on engagement in PA or exercise among adults living with HIV. We conducted a scoping review using the Arksey and O’Malley Framework. We searched databases between 1996 and 2021. We included articles that examined PA or exercise among adults with HIV and addressed at least one SDOH from the Public Health Agency of Canada Framework. We extracted data from included articles onto a data extraction charting form, and collated results using content analytical techniques. Of the 11,060 citations, we included 41 articles, with 35 studies involving primary data collection 23 (66%) quantitative, 8 (23%) qualitative, and four (11%) mixed methods. Of the 14,835 participants, 6398 (43%) were women. Gender (*n* = 24 articles), social support (*n* = 15), and income and social status (*n* = 14) were the most commonly reported SDOH in the literature with the majority of studies addressing only one SDOH. Future research should consider the intersection between multiple SDOH to better understand their combined impact on engagement in PA or exercise among PLWH.

## 1. Introduction

In Canada, approximately 62,050 people were living with HIV (PLWH) in 2018, with an incidence of 2242 new cases [1]. HIV prevalence has been increasing in Canada, due to a combination of increased incidence and lengthened life expectancy due to the advent of antiretroviral therapy (ART) [2,3]. As individuals are living longer, HIV is considered a chronic condition [4] characterized by health consequences related to HIV, aging and other health conditions, such as mental health issues [5], cardiovascular disease [6], opportunistic infections [7], and musculoskeletal disorders [8]. These health-related consequences can be defined as disability, including physical, cognitive, mental and emotional symptoms and impairments, which add challenges to carrying out daily activities, social inclusion, and worrying about future health and uncertainty [2,9,10,11].

Physical activity (PA) and exercise are an effective rehabilitation strategy for addressing disability and improving health outcomes among PLWH [12]. PA is defined as incorporating any bodily movements which creates energy expenditure, including conditioning exercises, household chores, engagement in sports and other activities which use movement, whereas exercise is defined as a subset of PA, which is planned, structured and repetitive with the goal of improvement or maintenance of physical fitness [13]. Engagement in PA of moderate to vigorous intensity is correlated with better health outcomes in the general population [14]. Systematic review evidence indicates that engagement in exercise at least 3 times per week for at least 4 weeks is safe and can lead to benefits in cardiorespiratory, strength, weight and body composition and quality of life outcomes among PLWH [15,16].

The World Health Organization (WHO) recommends, adults aged 18–64 year, including those with chronic conditions or disability, should undertake at least 150–300 min of moderate-intensity aerobic PA, or at least 75–150 min of vigorous-intensity aerobic PA, or an equivalent combination of moderate-intensity and vigorous-intensity activity throughout the week with at least 2 days of muscle strengthening activities to obtain benefits of engaging in PA and exercise [17]. For older adults, the guidelines include additional emphasis on enhancing balance and preventing falls [17]. Despite recommendations and documented benefits, engagement in PA and exercise varies among PLWH. Systematic review evidence suggests that rates of physical inactivity among PLWH ranged from 19–73%, indicating a range of exercise uptake among the population [18].

The role of social determinants of health (SDOH) on engagement in PA and exercise among PLWH is emerging. Social determinants of health (SDOH) are defined as conditions with which people are born, grow, live, work and age, that shape their level of power, income, and other determinants of life [19]. These determinants can play a crucial role in creating health inequities, which are avoidable differences in our health status [19]. Among PLWH, determinants such as cost associated with exercise, health status and disability along with neighbourhood and vulnerability to stigma influenced engagement in PA and exercise among PLWH [20,21]. Other barriers to exercise included parenting and employment responsibilities, attributed to relationships PLWH have with gender roles, education level, and socioeconomic status. In a review of PLWH, the role of social support and socioeconomic status influenced ability and level of engagement in exercise [22]. Although previous studies have examined the role of specific independent SDOH on engagement in PA and exercise, a comprehensive review of the collective role of SDOH on exercise in the context of HIV is unknown. Our aim was to characterize the literature pertaining to the role of SDOH on engagement in PA or exercise among adults living with HIV.

## 2. Materials and Methods

We conducted a scoping review to investigate what is known and not known about SDOH and engagement in PA or exercise among adults (ages 18 and over) living with HIV.

*Conceptual Foundation:* We used the Public Health Agency of Canada (PHAC) SDOH framework as a conceptual foundation for our review. This framework encompasses 12 determinants: income and social status, employment and working conditions, education and literacy, childhood experiences, physical environments, social supports and coping skills, healthy behaviours, access to health services, biology and genetic endowment, gender, culture, and race/racism [23] (Appendix A). We included any articles that referenced PA or exercise regardless of definition, that reported any form of bodily movement produced by skeletal muscles that requires energy expenditure [13].

We used the PRISMA-ScR (Preferred Reporting Items for Systematic reviews and Meta-Analyses extension for Scoping Reviews) [24], Joanna Briggs Institute Manual for Evidence Synthesis [25], and Arksey and O’Malley Framework, supplemented by Levac, Colquhoun & O’Brien to inform the methodology of this review [26,27].


**Step 1: Identifying the Research Question**


We set out to answer the following research question: what is the nature and extent of evidence related to the relationship of SDOH on engagement in PA or exercise among adults living with HIV? We defined *nature of evidence* as the type of literature on the relationships between SDOH and in engagement of PA or exercise among adults living with HIV (e.g., study location, study design, relationship of SDOH and engagement in PA or exercise among adults living with HIV). We defined *extent of evidence* as the amount of literature on the relationships between SDOH and engagement in PA or exercise and their role with adults living with HIV (e.g., number of publications).


**Step 2: Identifying Relevant Studies**


We searched Medline, CINAHL, Embase, PsycInfo, SPORTDiscus and Web of Science to identify research studies pertaining to SDOH, PA or exercise and HIV. We used a two concept Medline search on PA or exercise and HIV, using both a combination of keywords and MeSH headings and validated search filter with modifications [28]. Supplementary search terms were added to capture research relevant to the PHAC framework on childhood experiences, healthy behaviors, biology and genetic endowment. We adapted the final Medline search for syntax and subject headings for the other databases. See Appendix A for the Medline search strategy.

We included articles that addressed (i) PA or exercise, (ii) adults living with HIV (18 years and older) and (iii) addressed at least one SDOH as conceptualized by the PHAC SDOH Framework. We included articles published from 1996 and onwards and were published in English language. We limited our search to 1996 onwards, as this is when Highly Active Antiretroviral Therapy (HAART) emerged resulting in prolonged survival of PLWH [29].


**Step 3: Study selection**


We imported citations (titles and abstracts) yielded from the search into EndNote referencing software Version X9 [30] (Clarivate: Philadelphia, PA, USA) and then into DistillerSR software Version 2.35 [31] (DistillerSR Inc., Ottawa, ON, Canada) to facilitate the process for screening citations and abstracts for study selection. We reviewed studies for inclusion in three stages, title review, abstract review, and full-text review.

*Title review:* Two reviewers (FS, NM) independently screened each title for the ability to address HIV, PA or exercise and SDOH by answering the following question: “*Is the focus of the article to examine, explore, measure (or refer to) the associations of SDOH with PA and/or exercise?*” with “Yes”, “No” and “Unsure” options. If any two of three components (HIV, SDOH, and PA or exercise) were included in the title, reviewers responded “Yes”. The article was excluded if both reviewers responded “No” to the above-mentioned question. If a reviewer responded “Unsure”, the article was included in the subsequent stage for abstract review.

*Abstract review:* The same two reviewers (FS, NM) piloted the study selection criteria with five initial articles by independently screening five abstracts and answering the following questions, based on the inclusion criteria:Does the article include (or refer to) adults living with HIV? (Yes, No, Unsure)Does this article include (or refer to) PA and/or exercise? (Yes, No, Unsure)Was this article published from 1996 and onwards? (Yes, No, Unsure)Is the primary focus of the article to examine, explore, measure (or refer to) the associations of the SDOH (income and social status, education and literacy, employment and working conditions, physical environments, healthy behaviors, social support and coping skills, childhood experiences, access to health services, biology and genetic endowment, gender: culture, and race/racism) with PA and/or exercise? (Yes, No, Unsure)

All authors (SR, KOB, FS, NM) conducted a second abstract piloting process of 10 additional citations. The study team met four times to discuss and refine the abstract review process. Two reviewers (FS and NM) independently completed the abstract review. If both reviewers responded “Yes” to all of four inclusion questions, these articles were included in the full article review. If both reviewers responded “No” to any of the four inclusion questions, then the article was excluded. If any of the responses were “Unsure” by at least one of the reviewers for an article, the article was included in the subsequent stage of full-text review.

*Full-text review:* Two reviewers (FS and NM) independently reviewed full text of articles to determine inclusion. We piloted the process with 10 articles involving the four questions above for inclusion. All authors met to discuss the findings from the full text review. If any uncertainty arose pertaining to article inclusion, a third reviewer (KKO or SR) determined the final inclusion. Two authors (FS and KKO) contacted authors if we could not retrieve a full-text article or if data specific to PLWH were not included in the article. We refined our study selection criteria to include articles with aims or objectives that directly related to examining the relationship of SDOH on PA or exercise among PLWH (opposed to articles that did not a priori consider SDOH in their aims or objectives, but may have discussed SDOH as an emergent finding).


**Step 4: Charting the Data**


We extracted data from included studies for the following variables: title, author(s), year of publication, study location, city and country of lead author, study purpose, study objectives, study design, type of article (e.g., primary data collection, review article, etc.), data collection method, study participant characteristics (including number and proportion of women), types of outcomes measured, intervention (if applicable), conceptual framework used (if applicable), SDOH reported in the article, authors’ results and conclusions, authors’ results and conclusion specific to PA and SDOH, our reviewer interpretations, and additional notes. Two reviewers (FS and NM) extracted the data from included studies using a data charting form on Microsoft Excel [32] (Microsoft, Redmond, WA, USA). See Appendix A for the operational definitions of concepts extracted from the included articles.

We piloted the data extraction form in conducted two rounds. Two reviewers (FS and NM) independently extracted data from five included articles using the data extraction form to ensure consistency, followed by an additional five articles. Following each round of the pilot, the entire team met to refine the data charting form. Two reviewers extracted data from the 10 pilot articles using the data extraction form, which was reviewed by the team to ensure comprehensiveness and consistency in the data extraction process.


**Step 5: Collating, summarizing and reporting the results**


We described characteristics of included articles using frequencies and percent for categorical variables and content analytical techniques for text data (e.g., author’s results and conclusions related to PA and SDOH) [25,33,34,35,36,37,38]. The research team met four times throughout this step to discuss our synthesis and reporting of results.

## 3. Results

We yielded 15,836 citations from the search strategy from CINHAL (*n* = 2690), Embase (*n* = 3463), Medline (*n* = 3423), Web of Sciences (*n* = 4641), PsycInfo (*n* = 1326), and SportDiscus (*n* = 293) databases. After removing 4754 duplicates, this resulted in 11,060 citations. After screening title and abstracts, we retained 89 articles for full-text review. Of the 89 full-text articles reviewed, 41 articles were included in the review. See Figure 1 for the PRISMA Diagram. See Appendix A for the citations of excluded studies classified as indirectly related to SDOH.

### 3.1. Characteristics of Included Studies

Table 1 describes the characteristics of the included articles. Among the 41 included articles, 35 (85%) were research studies involving primary data collection [21,34,35,36,37,38,39,40,41,42,43,44,45,46,47,48,49,50,51,52,53,54,55,56,57,58,59,60,61,62,63,64,65,66,67] one (2%) was a secondary analysis [68], three (7%) were systematic reviews [69,70,71], and two (5%) were narrative reviews [22,72] (Figure 1).

*Study Design:* Among the 35 studies involving primary data collection, 23 (66%) were quantitative [34,35,37,38,40,42,44,45,48,49,50,51,52,53,54,57,59,61,62,63,64,65,66], eight (23%) qualitative [21,36,39,41,47,55,56,58], and four (11%) mixed method studies [43,46,60,67]. Among the 23 quantitative studies, 18 (78%) studies were cross-sectional [34,37,38,40,42,44,45,48,49,50,52,53,54,59,62,63,64,66], two (9%) cohort studies [35,61], two (9%) intervention studies [51,57], and one (4%) was a randomized controlled trial (RCT) [65].

*Data Collection:* Four (50%) of the qualitative studies involved focus groups [36,39,55,56] and the remaining four involved semi structured interviews for data collection [21,41,47,58]. Quantitative studies used questionnaires, clinical records, medical examination, and different PA measurement tools.

*Study Location:* Among the 35 studies involving primary data collection, study locations included the United States (*n* = 10; 28%) [35,36,39,40,49,55,56,59,64,65], South Africa (*n* = 5; 14%) [41,50,51,52,60], Uganda (*n* = 3; 8%) [38,63,67], Malawi (*n* = 2; 6%) [34,53], Canada (*n* = 2; 6%) [21,58], and the UK [57], Vietnam [42], Rwanda [45], China [48], France [47], Brazil [62], Thailand [54], Switzerland [61], Columbia [46], and Nigeria [44] one study each (3%). Three studies (8%) involved more than one study location [37,43,66].

*Date of Publication:* The publication date of included articles ranged between the year 2004 and 2021. Figure 2 displays the timing of publications, which represents the trend for consideration of SDOH in the context of HIV and PA or exercise over the years.

*Primary Purpose of Included Articles*: Thirty-one (76%) of the 41 included articles included a primary purpose to explore the association or relationship of SDOH with PA or exercise in PLWH [21,22,36,38,39,40,41,42,43,44,45,47,51,52,53,54,55,56,57,58,59,60,62,64,66,67,68,69,70,71,72], whereas 10 (24%) articles examined SDOH in the context of other aims [34,35,37,46,48,49,50,61,63,65]. Among these 10 articles, four (40%) evaluated the association of SDOH with PA or exercise as a secondary aim of the study [46,49,61,63]. Among the three systematic reviews, one (33%) determined the correlates of PA in PLWH by reviewing six studies [71], and the remaining two (67%) systematic reviews investigated the prevalence and predictors of dropout in PA interventions by reviewing 36 studies [69], and evaluated barriers and facilitators of PA participation in PLWH by synthesizing 45 studies [70]. Among the two narrative reviews included in this review, their purposes were to discuss the opportunities and challenges of benefiting from different types and effects of PA and to examine the literature on PA, social support and socioeconomic status (SES) and to generate recommendations for designing and implementing PA interventions with PLWH [22,72].

### 3.2. Characteristics of Participants

In this section we describe characteristics among studies involving primary data collection. Two studies included the same study population within the same study, hence we report characteristics of the sample from one article only [38,63] resulting in a total of 34 research studies in this review. See Table 1 for characteristics of participants in included studies.

Among the 14,835 participants in the 34 research studies, sample sizes ranged between 8 and 8104 participants. The study population were PLWH in all but one study, where 30 out of 59 participants (51%) were HIV positive and the remainder were HIV negative [67]. Among the 34 studies, four (12%) included women only [37,43,51,59] and one (3%) study included men only [55]. Of the total sample of participants in the 34 studies (*n* = 14,835), 57% (8437) were men and 43% (6398) were women.

Terminology and definitions of PA and exercise varied across the 35 studies involving primary data collection. Eighteen studies (51%) did not provide a definition of PA or exercise [39,43,46,48,49,50,51,52,55,56,57,58,59,60,61,64,65,66]. Among the remaining 17 studies, 13 (76%) referred to the definition of PA and four (24%) used a definition of exercise. Among the 13 studies that referred to the definition of PA, 7 (54%) followed recommended guidelines of PA by the World Health Organization (WHO), which was a total of at least 30 min of moderate-intensity aerobic PA per day, five or more days a week; or three or more times per week for at least 20 min of vigorous-intensity aerobic activity [34,42,44,45,47,53,63]. Exercise definition varied in four studies [21,36,41,54].

### 3.3. Evidence Related to Relationship of Social Determinants of Health with Physical Activity and Exercise

Nine out of 12 SDOH were addressed across the 41 included articles. The number of SDOH addressed in each article ranged from one [34,37,39,40,43,44,45,49,50,53,56,64,65,66] to eight [70] (Table 2). The most common SDOH addressed across included articles was gender (*n* = 24; 47%), followed by social support and coping skills (*n* = 15; 36%), income and social status (*n* = 14; 34%), education and literacy (*n* = 9; 22%), employment and working conditions (*n* = 9; 22%), physical environment (*n* = 9; 22%), healthy behaviors (*n* = 9; 22%), race/racism (*n* = 6; 15%), and culture (*n* = 5; 12%). (Table 2; Figure 3). Three SDOHs (childhood experiences, access to healthcare, and biology and genetic endowment) were not addressed in any of the included articles. None of the articles utilized a pre-determined SDOH framework to report their findings.

Given the heterogeneity of the study aims, study design and SDOH addressed across the articles, we describe results of the individual articles for each of the SDOH below.

#### 3.3.1. Gender

Gender was reported among 24 (58%) of the 41 articles with mixed results [34,35,37,42,44,45,46,48,50,52,53,54,57,59,61,62,64,65,66,67,68,69,70,71]. Several cross-sectional studies reported that on average, women living with HIV engaged in lower levels of PA than men living with HIV. For example, Chisati et al. (2020) reported 51% of women participants had low PA levels compared to 22% of men with HIV with less than half of women engaged in high intensity PA compared to men (16.7% vs. 37%) [34]. One cross-sectional study collected PA data by using the Global Physical Activity Questionnaire (GPAQ) and found men living with HIV engaged in more PA (overall PA score on GPAQ 480.3 min/week) than women (overall PA score 269.1 min/week) in work-related, transport-related, and leisure-related PA, but especially in work-related PA (men: 279.3 min/week, women: 125.9 min/week) [52]. Additionally, another study reported women had higher odds of not adhering to PA recommendations compared with men (Odds Ratio (OR) 1.62; 95% confidence interval (CI): 1.01 to 2.57) [38]. Muronya et al. (2011) reported greater physical inactivity among women than men (28% vs. 25%); however, gender was not statistically associated with PA [53]. A cross-sectional study found that women living with HIV engaged in less PA than men, but only during middle adulthood (36–50 years) (women: 2.4 h of exercise/ week, men: 4.5 h/week *p* < 0.05) [64]. Two cohort studies reported similar findings, where men were more physically active than women [35,61]. One RCT aimed to evaluate the 3- and 6-month effects of an intervention (System CHANGE) on PA and dietary quality in PLWH at high risk of developing cardiovascular diseases (CVD) and reported women were consistently associated with engaging in less PA compared with men [65]. Wright et al. (2020) found that men engaged in more intentional PA, but women engaged in more PA related to activities of daily living, indicating that the definition of and type of PA was correlated with gender differences [67]. One study found a low level of PA among women from low SES, with limited time spent in vigorous activities among groups of women from Rio Di Janeiro (3 min/day on average) and Maputo (5 min/day on average), with daily light PA higher among women living in Rio (111 min/day) compared with Maputo (84 min/day). However, data were not compared to PA among men [37]. Dang et al. (2018) was the only study to find that women gender was weakly associated with higher PA levels as measured by the International Physical Activity Questionnaire (IPAQ) (correlation coefficient 0.25; 95% CI: 0.11, 0.39) [42]. Several studies found no significant associations between gender and PA level [44,50,57]. Two systematic reviews could not conclude whether men or women engaged in more PA due to inconsistent findings among included studies [70,71]. Another systematic review and meta-analysis investigated the prevalence and predictors of treatment dropout in PA interventions in PLWH and reported a lower proportion of men participants (ß = 1.15, SE = 0.49, z = 2.0, *p* = 0.048) moderated the higher dropout rates [69].

#### 3.3.2. Social Support and Coping Skills

Social support and coping skills, and PA among PLWH were reported in 15 (37%) of the 41 articles [21,22,36,38,40,41,47,49,51,55,58,59,60,63,70]. Fourteen of the 15 studies found that social support was an important facilitator of PA among PLWH. Gray et al. (2019) conducted a qualitative study with active and inactive PLWH in which they found a lack of social support to be a barrier in engaging in PA [47]. Another qualitative study conducted in Canada reported social influence, such as encouragement from friends and health care providers, as a facilitator to PA [58]. A narrative review and a qualitative study reported that having an exercise partner or encouragement from staff members at the community facilities, such as churches or community centres increased motivation and adherence to PA [21,22]. Similarly, another qualitative study reported that encouragement received from families and community members helped participants adhere to exercise, and, in many cases, encouraged their children, partners, and neighbors to start exercising [41]. Authors reported that support from family members was a facilitator for exercise. For instance, a cross-sectional study measuring benefits and barriers to PA among PLWH in the United States showed that not receiving encouragement from a spouse, significant other, or family member was a perceived barrier to engagement in PA [59].

The social setting where PLWH were engaging in PA was also important. Mabweazara et al. (2018) found that community/group exercise settings helped PLWH stay motivated within a cost-effective environment that facilitated social-bond formation [22]. This was especially useful among PLWH since stigma often led to lower levels of social support. Other studies found social support to be an important facilitator to engagement in PA, including Neff et al. (2019) who reported “*social support was an important factor for motivating and continuing exercise*” [55]; Ley et al. (2015) reported “*the need for social support or peer support, but essentially trustful and confidential support, from a good friend*” [51]; and Roos et al. (2015) reported “*an important facilitator was the support and motivation received from friends and family through participating in sessions and walking with participants*” [60]. In summary, social support including encouragement from friends, family members, staff at the gym, and peers in group exercise all appeared to be important facilitators to engaging in PA among PLWH.

#### 3.3.3. Income and Social Status

Fourteen (34%) of the 41 articles examined relationships between income and social status on PA among PLWH [21,22,36,39,41,47,51,55,58,59,62,67,70,72]. All 14 articles reported that financial constraints or costs were a barrier to participation in PA among PLWH, within different contexts and geographic locations. For example, a qualitative study among older exerciser and non-exerciser groups reported both groups identified costs as a barrier to exercise [36]. Similarly, two qualitative studies among older adults living with HIV found cost was a barrier to PA [55,58]. A mixed method study found expense as a barrier to exercise in both HIV positive and negative groups [67]. Two of the studies focused on women from disadvantaged urban settings in South Africa [51] and from the southern United States [59] and reported similar results. Examples of financial barriers were costs of a gym membership [39] and transportation to PA facilities [51]. Financial insecurity also led to food scarcity, which was a barrier to exercise among PLWH in Kwazulu-Natal South Africa [41]. A systematic review incorporated one study in their review that highlighted the association between higher income and higher PA [70].

#### 3.3.4. Education and Literacy

Thirteen (32%) of the 41 articles reported on education and literacy, and PA or exercise among PLWH with varying results [35,38,42,46,48,52,54,61,62,63,68,70,72]. For example, authors of a cohort study reported PA levels did not correlate with years of education (r = 0.09, *p* = 0.61) [35]. Mabweazara et al. (2018) reported education was not a significant predictor (*p* = 0.057) of overall PA among PLWH [22], and Dang et al. (2018) found that high school education did not differ regarding PA compared to PLWH with lower levels of education (OR 1.52, 95% CI: 0.94, 2.46) [42]. Another study reported no significant difference in PA stages of change related to education level [63]. Hsieh et al. (2014) found higher levels of PA were significantly associated with lower education (OR: 0.50, 95% CI: 0.27, 0.91) among Chinese individuals living with HIV [48]. Alternatively, some studies found the association between lower levels of education and lower PA level among PLWH. For instance, a study reported participants who never attended school were less likely to engage in physical exercise compared to those who had secondary or higher education levels (Adjusted Odds Ratio (AOR): 0.22; 95% CI: 0.08, 0.55; *p* = 0.001) [54]. Similarly, another study found low education (up to 4 years of study, prevalence ratio 1.71) was associated with physical inactivity [62]. A narrative review conducted in South Africa highlighted physical inactivity and obesity were strongly related to lower education level [72]. Nevertheless, a few studies found an association between higher education and higher levels of PA. Mabweazara et al. (2021) showed that educational attainment (β = 0.127; *p* < 0.001) significantly predicted total moderate-to-vigorous PA among PLWH of low SES in Cape Town, South Africa [68]. Another study reported, participants with higher education reported more free-time PA than those with lower education among PLWH from the Swiss HIV Cohort Study [61]. A mixed methods study examining patients’ preferences for HIV treatment in Colombia found higher educated patients identified PA as the most important HIV treatment attribute, whereas low educated patients valued accessibility to clinic over PA [46].

#### 3.3.5. Employment and Working Conditions

Nine (22%) of the 41 articles reported on the relationship between employment and working conditions and PA among PLWH [38,42,48,52,60,63,68,69,70]. Results from these individual studies suggest that employment was associated with engagement in PA among PLWH, with higher levels of PA and exercise among those in non-sedentary jobs. A cross-sectional study showed that higher levels of PA was significantly associated with a higher likelihood of manual labor versus non-manual labor occupation (*p* = 0.002) [48]. Another cross-sectional study reported unemployed PLWH had a 2.81 (95% CI: 2.00, 3.94) higher odds of not adhering with PA recommendations compared with employed PLWH [38]. A cross-sectional study found that blue-collar workers or farmers (OR 2.24; 95% CI: 1.27, 3.95) were more likely to have a higher International Physical Activity Questionnaire (IPAQ) score and were classified as physically active compared with white-collar workers [42]. Similarly, a RCT found that sedentary jobs prevented PLWH from achieving adequate step counts, as well as following their PA programme outside of work due to fatigue and busy schedule among PLWH in South Africa [60]. A cross-sectional study reported employment was a significant predictor of overall PA [22]. Conversely, a cross-sectional study found no difference in employment status between the different stages of change in regard to PA [63]. One meta-analysis revealed dropout rates in PA interventions in PLWH were not moderated by employment status (*β* = −1.81, 95% CI, −5.69, 2.06; *p* = 0.36) [69].

#### 3.3.6. Physical Environments

Thirteen (32%) of the 41 articles reported on the relationship between physical environment and PA or exercise among PLWH [21,36,47,51,55,56,58,59,60,67,70,71,72]. Physical environment often was described as either a barrier or facilitator to participating in PA. A qualitative analysis described how physical environmental factors such as gym culture, feeling unsafe in their neighborhood, or lack of affordable housing were barriers to PA among PLWH [36]. Nguyen et al. (2017) also found in their qualitative inquiry that perceived safety or culture in the gym environment was a barrier, specifically pertaining to exercising in “non-positive” spaces due to fear of stigma [56]. Another qualitative study reported HIV-positive exercise programs were a facilitator to PA, by creating a safe and inclusive environment [21]. A mixed methods intervention study reported a lack of safety and security in disadvantaged areas as a barrier to PA among South African women living with HIV [51]. Another mixed methods study found safety concerns was a common barrier to PA among PLWH in Uganda [67]. In their narrative review, Ley and Barrio (2012) reported that limited space and desolate living conditions failed to offer suitable environments to engage in PA in South Africa [72]. Two qualitative and one mixed method studies found that climate constraints were a barrier to engaging in PA [47,58,60]. Lastly, a qualitative study among older PLWH reported that convenience of location to an exercise facility was a motivating factor to initiate exercise [55]. While the SDOH “physical environments” can refer to many different concepts, common themes were gym culture, safety, and climate.

#### 3.3.7. Healthy Behaviors

Nine (22%) of the 41 articles reported on healthy/unhealthy behaviors in relation to PA among PLWH that largely concluded smoking, and substance use did not have an influence on engagement in PA [36,38,48,54,61,62,63,69,70]. We did not include “physical activity” as a healthy behavior since it was our outcome of interest. Hsieh et al. (2014) found that smoking (OR: 1.49, 95% CI: 0.81, 2.77), and alcohol use (OR: 1.56, 95% CI: 0.75, 3.21) were not significantly related with PA [48]. Similar findings were reported in a cross-sectional study where no association was found between smoking (*p* = 0.06) and alcohol use (*p* = 0.15) and adherence to PA [38]. One qualitative study reported drug use as a barrier to PA and exercise [36]. Only one systematic review examined drop out rates among PLWH engaged in PA or exercise and reported smoking status did not moderate dropouts rate (*β* = −1.78, 95% CI, −3.67, 0.11, *p* = 0.07) [69].

#### 3.3.8. Culture

Five (12%) of the 41 articles reported on culture with PA or exercise among PLWH [41,43,51,71,72]. A narrative review and one mixed methods intervention study examined the cultural barriers to exercise among women in the Black African community [51,72]. “*In the disadvantaged community, for example, women generally are not seen running in the street…due to social-cultural norms and attitudes…*” [51]. A mixed method study also discussed cultural perceptions of exercise as a barrier for PLWH engaging in PA: “*many participants were apprehensive about exercise in terms of their cultural practices and were not sure if this was something they would really do*” [43]. Finally, a qualitative study examined stigma as a cultural factor as a barrier to engaging in PA [41].

#### 3.3.9. Race/Racism

Six (15%) of the 41 articles reported on race and racism among PLWH with mixed results [35,57,62,69,70,72]. Two studies showed no race difference in PA. For instance, a longitudinal study in the United States found PA did not differ by White vs. non-White PLWH [*t* = 0.12, *p* = 0.91] [35]. In a systematic review, authors found that the dropout rate from PA programs was not moderated by race or ethnicity among PLWH [69]. Authors of a narrative review found Black women living with HIV experienced barriers to engaging in PA due to cultural perceptions, addressing the intersection between gender, race and culture [72]. An observational intervention study reported the proportion of Black African PLWH among a group non-adherent with PA (73% vs. 9% White British) and the White British (46% vs. 27% Black African) in a group adherent to PA were significantly higher. Authors suggested further investigation on the relationship between race and PA among PLWH [57]. In a systematic review, authors could not ascertain a relationship between race and PA among PLWH among the included studies as one study showed higher PA among the non-White PLWH, while another showed the opposite and four studies showed no association between race and engagement in PA or exercise [70].

## 4. Discussion

To our knowledge, this is the first scoping review that examined the nature and extent of evidence on the relationship between SDOH and engagement in PA or exercise among adults living with HIV. Our findings indicate that gender was the most common SDOH among included articles, followed by social support and coping skills, and income and social status. Most included articles (76%) explored the relationship between SDOH and PA or exercise in PLWH as a primary aim, whereas the rest of the articles (24%) reported SDOH as a secondary aim. Among the 23 quantitative studies, the majority (78%) used a cross-sectional design that prevented establishing a causal relationship between the SDOH of interest and engagement in PA or exercise. Eight qualitative studies highlighted socio-environmental factors as barriers and facilitators of PA or exercise, with financial cost, social support, cultural context, and physical environment (e.g., gym environment, safe place, and weather) as the most common SDOH influencing engagement in PA or exercise among PLWH.

Despite the explicit focus of articles examining relationships between different SDOH and PA or exercise among PLWH, none of the articles used a SDOH framework to inform their analytical approach. In addition, a large number of studies examined only a single SDOH [34,37,39,40,43,44,45,49,50,53,56,64,65,66], while only one systematic review evaluated eight SDOHs identified in the PHAC framework [70]. Identifying more than one SDOH is important to evaluate the intersection between different determinants. Although, some studies evaluated more than one SDOH in relation to PA or exercise (Table 2), separate results were reported for each determinant. In two studies, women from low SES [37] and Black African women from disadvantaged communities were found to engage in low PA or exercise [51]. None of the other studies appeared to report the impact of intersectionality on PA or exercise. This scoping review indicates an evidence gap in the literature as childhood experiences, biology and genetic endowment, and access to health services were not reported in the included articles, highlighting the need for future research to address these determinants and and the impact of combinations of SDOH among PLWH engaging in PA or exercise.

Results from individual studies in this scoping review suggest that women living with HIV engage in less PA compared to men living with HIV. Only one study reported higher PA among women [42], whereas other studies reported non-significant findings. However, women were underrepresented in most of the included studies. Gender remains an important SDOH to consider in the context of PA and exercise among PLWH. While gender was the most common SDOH addressed in the review, genders other than men and women were underrepresented. Only five studies included other genders [46,49,56,65,66], among which one study reported combined female and transgender data [56] (Table 1). Moreover, some studies utilized sex and gender interchangeably. Transgender groups are at higher risk of HIV, specifically transgender women have 49 times the odds of having HIV compared to the general population [73]. Thus, future studies should consider other genders in addition to men and women on engagement in PA or exercise among PLWH.

Our review suggests no association between unhealthy behaviours such as smoking and alcohol use with engagement in PA. Smoking and alcohol consumption are not considered as healthy behaviors because of their adverse impacts on physical and mental health [74,75]. Interestingly, a cross-sectional study from Brazil reported a higher level of PA among young, middle aged and older adults who were weekly alcohol consumers [76]. We recommend evaluating the relationship between healthy or unhealthy behaviours and PA or exercise in larger population-based studies involving PLWH.

Our findings suggest that stigma associated with culture and race/racism were underrepresented in the literature, with only 12% and 15% of included studies addressing these SDOH, respectively. While one systematic review was inconclusive regarding the relationship between race and PA [70], authors of one narrative review suggested engagement in PA was lower among Black women due to perceived barriers [72]. Some literature was suggestive of an association between White race with higher PA in general population. For example, one study conducted a secondary data analysis from the National Health and Nutrition Examination Survey of 9472 adolescent and young adult respondents from 2007 through 2016 reported White race and higher income were associated with greater PA in adjusted models [77]. The Centre for Disease Control and Prevention (CDC) conducted a Behavioral Risk Factor Surveillance System (BRFSS) among 52 United States jurisdictions from 2017 to 2020, and showed lower prevalence of physical inactivity outside of work among non-Hispanic White (23%) compared to non-Hispanic Black (30%) participants [78]. Based on the 2003–2009 American Time Use Surveys (ATUS) data, Saffer et al. (2019) showed non-work PA was significantly lower among Blacks relative to non-Hispanic Whites [79]. Conversely, total work related PA was significantly higher among Black peoples compared to White and Asian peoples given the context that Black individuals were more likely to work in blue-collar and physically demanding occupations [80]. Although these findings involve the general population, the association between race and PA engagement among PLWH needs further evaluation. We recommend examining the intersection between race, gender, employment, education, and culture among PLWH to better understand engagement in PA or exercise.

This review identified gaps in the literature related to SDOH and engagement in PA or exercise among HIV. Definitions of PA and exercise varied significantly across the studies. None of the included studies adopted a SDOH framework as the theoretical basis for their work. Additionally, authors used a variety of self-reported questionnaires and objective measures of PA and exercise. Considering the diversity in PA definition and assessments, and benefits and limitations of objective versus self-report of PA, researchers should consider the inclusion of an operational definition of PA or exercise in future work and provide a rationale for the selection of the PA or exercise assessment tools. Similarly, authors examining SDOH should better define the determinant(s) of interest and the research context. Clearly defined PA, exercise and SDOH variables using a conceptual framework to guide the analytical work and standardized and validated outcome measures should improve the interpretation of evidence in this field and foster standardized approaches in future research. Furthermore, the majority of studies were cross-sectional in nature making it difficult to identify the causal relationship between SDOH and PA or exericse. Future longitudinal or cohort studies among PLWH, may help to further understand the strength and direction of the relationships and whether PA or exercise is a moderator or mediator of health outcomes among PLWH. Another major gap in the evidence is the absence of research on the impacts of the intersection between multiple SDOH on PA and exercise. Finally, many studies included small sample sizes with unclear transferability to the broader HIV population, highlighting the need for a richer understanding of the collective influence that a combination of SDOHs have on engaging in PA or exercise among PLWH.

Other research has explored the relationship between SDOH and physical activity or exercise among other chronic illness populations. This body of work has also documented that the ability for individuals to engage in exercise is affected by cost, health status and disability [80]. Social determinants research in other chronic conditions found similar barriers and facilitators of engagement in PA. For example, people living with diabetes with a low socioeconomic status and lower education level, found it difficult to engage in PA regularly [81]. In a study conducted with adults living with multiple chronic conditions, an unsafe environment, which attributes to low socioeconomic status and housing were correlated with lower engagement in PA [82]. One qualitative study among people with type 1 diabetes who progressed to end-stage renal disease identified lower socioeconomic status, lower education level, and female gender as barriers to PA [83]. Social support and inclusion were also determinants and found to facilitate and encourage engagement in adults living with chronic conditions [84].

Strengths of this scoping review includes our rigorous adherence to well-established scoping review methodology [24,25,27], our broad selection criteria involving a range of study designs and methodologies, and our team based approach involving independent review and multiple piloting of the inclusion and data extraction process. We did not assess quality of the included studies, as this is not a requirement of scoping reviews but rather to provide an overview of the evidence [25]. Potential limitations of our review is that we may have missed relevant articles despite our extensive search strategy of multiple databases. We contacted the corresponding authors of titles and abstracts without full-text articles and waited up to 4 weeks before excluding them. We only included English language articles because of the cost and time associated in translating foreign materials. Despite these limitations, this review provides important insight into the nature of extent of evidence on the role of SDOH on engagement in PA or exercise among PLWH.

## 5. Conclusions

This scoping review characterized the evidence examining the role of SDOH on PA or exercise among PLWH. Gender was the most commonly reported SDOH in the literature with the majority of studies addressing only one SDOH. Results from individual studies suggest that social support is a facilitator, and financial constraints and costs are a barrier to engagement in PA or exercise among PLWH. Results of this review may help to inform clinicians, researchers, and policy makers to better understand the role of SDOH as potential barriers or facilitators while promoting or evaluating PA or exercise with PLWH. Future research should consider the intersection between different determinants to better understand the combined impact of different determinants on engagement in PA or exercise.

## Figures and Tables

**Figure 1 ijerph-19-13528-f001:**
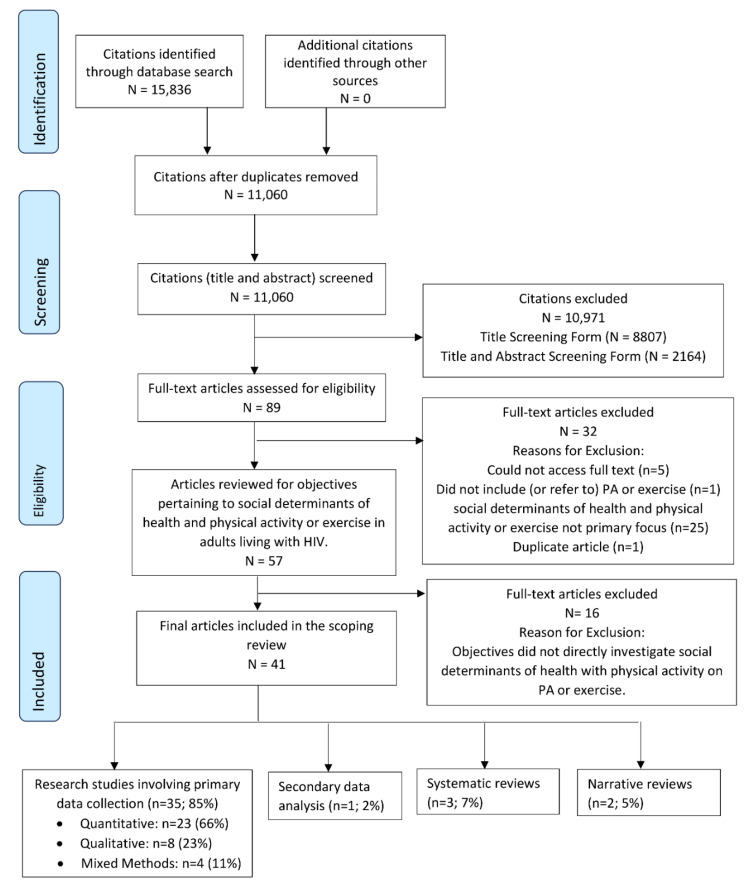
PRISMA Flow Diagram—Scoping Review: Evidence on Social Determinants of Health, Physical Activity or Exercise among Adults Living with HIV.

**Figure 2 ijerph-19-13528-f002:**
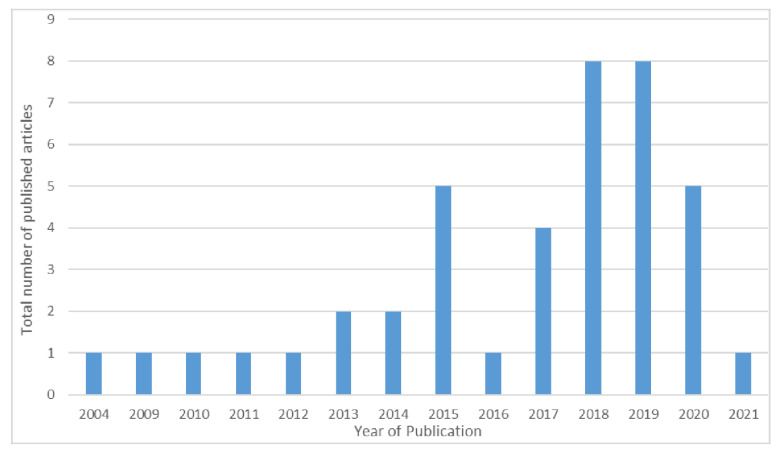
Timing of publications of the 41 included articles between the year 2004 and 2021.

**Figure 3 ijerph-19-13528-f003:**
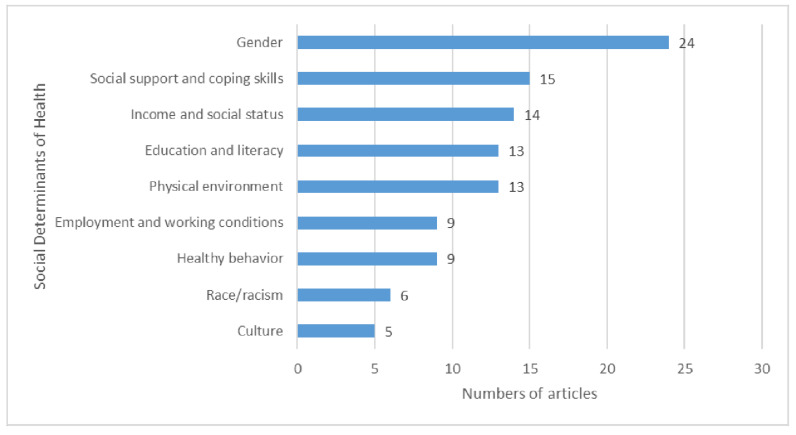
Social determinants of health (SDOH) represented across the 41 included articles examining physical activity or exercise among adults living with HIV.

**Table 1 ijerph-19-13528-t001:** Characteristics of included articles (*n* = 41).

First Author, Publication Year [Reference Number]	Study Setting Country	Study Design	Study Purpose	Sample Size; Gender (%) (Men (M), Women (W), Other (O)): Mean Age in Years (yrs) (SD) *	No. of Participants Living with HIV (%); No. of Participants Taking ART (%)	Physical Activity (PA) at Baseline # (%) of Participants Considered PA	Data Collection Method	Physical Activity (PA) or Exercise Definition	Authors Results/Conclusions Related to Social Determinants of Health (SDOH) and Physical Activity (PA)
Capili et al., 2014 [39]	United States	Qualitative	To explore the personal and in-depth detail of expectations, perceptions, and beliefs related to a healthy lifestyle and behavior in patients infected with HIV to identify the barriers and facilitators to engagement in lifestyle interventions.	*n* = 123 M: 74 (60%) W: 49 (40%) 48 yrs ± 7.3	123 (100%) NR	NR	Qualitative—focus groups	Not defined	Financial cost was cited as an inhibitory factor in undertaking certain types of physical activity among participants living with HIV; as one woman explained, ‘‘I would like to go to a gym but I don’t have the money for it.’’
Chisati et al., 2020 [34]	Malawi	Cross-sectional	To determine levels of PA among people living with HIV and receiving ART in Blantyre, Malawi.	*n* = 213 M: 81 (38%) W: 132 (62%) M: 37 yrs ± 6.7 W: 36 yrs ± 6.3	213 (100%) 213 (100%)	Low (40%), moderate (36%), high (24%) intensity level of PA	Questionnaire, Stadiometer	*PA*: Moderate level PA defined by 3 or more days of vigorous PA for at least 20 min per day or 5 or more days of walking or moderate intensity PA for at least 30 min per day; high level PA defined by at least 3 days of vigorous intensity PA accumulating at least 1500 MET minutes per week or 7 or more days of any combination of vigorous PA, moderate intensity PA or walking achieving a total of at least 3000 MET minutes per week.	A larger number of females (51%) had low PA levels compared to males (22%) living with HIV.
Cioe et al., 2019 [35]	United States	Cohort	To examine prospectively the impact of recommending increased daily PA on overall symptom burden and fatigue over a 12-week period in people living with HIV using a single-group within-participant design.	*n* = 40 M: 24 (60%) W: 16 (40%) 51.48 yrs ± 7.41	40 (100%) 40 (100%)	Range: 770 to 81,324 steps (mean 26,600 [SD ± 18,547]) at baseline	Questionnaire, Omron Tri-Axis Digital Pedometer	*PA*: 150 min of PA per week (30 min per day, 5 days per week).	At baseline, male participants walked significantly more steps per week (M= 31,882, SD = 20,439) than female participants (M = 18,501, SD = 11,684;, *p* = 0.02). At week 12, the gender difference in weekly mean step totals remained significant (*p* = 0.02), males 37,601 (SD = 28,607); females 16,386 (SD = 14,142) and did not differ by race (White vs. non-White; *p* = 0.91).
Clingerman, 2004 [40]	United States	Cross-sectional	To identify and explore relations among PA, social support, and health-related quality of life in persons with HIV who were living in community settings.	*n* = 78 M: 70 (90%) W: 8 (10%) 40.4 yrs ± 8.33	78 (100%) NR	NR	Questionnaires	Not defined	Weekly PA frequency and average friend social support explained 37.3% of the variance in health-related quality of life (*p* < 0.001). Standardized beta weights for PA frequency was 0.46, *t* = 4.04, *p* < 0.000; and for average friend support 0.337, *t* = 2.98, *p* < 0.01
Cobbing and Chetty, 2019 [41]	South Africa	Qualitative	To describe the experiences of people living with HIV involved in a novel home-based rehabilitation intervention in KWaZUlu-Natal, South Africa.	*n* = 8 M: 4 (50%) W: 4 (50%) Age Range: 23–41 yrs	8 (100%) NR	8 (100%)	Semi structured interviews	*Exercise*: Home based rehabilitation program including a combination of aerobic, strength, and functional exercises	The encouragement participants received from their families and community members helped them adhere to the exercises, and, they encouraged their children, partners, and neighbors to start exercising. Financial constraints limited access to institutional care and contributed to food scarcity, which affected full participation in the home-based rehabilitation intervention. An inhibitor to exercise was HIV stigma and, in some cases, additional discrimination associated with living with disability.
Dang et al., 2018 [42]	Vietnam	Cross-sectional	To determine the physical activity level and its associated factors among persons living with HIV receiving ART treatment.	*n* = 1133 M: 665 (59%) W: 468 (41%) 35.5 yrs ± 6.9	1133 (100%) 1133 (100%)	Minimally active: 181 (16%), health-enhancing physical activity (HEPA) active: 771 (68%)	Questionnaires	*PA*: Health-enhancing physical activity (HEPA) active -Vigorous activity for at least 3 days and obtained a total physical activity of at least 1500 MET-min/week, or 7 or more days of combination physical activities of walking, moderate-intensity or vigorous activities and obtained a total physical activity of at least 3000 MET-min/week; Minimally active: 3 or more days of vigorous activity of at least 20 min per day (800 MET-min/week), or 5 or more days of moderate activity or walking of at least 30 min per day, or 5 or more days of walking combining with moderate-intensity or vigorous-intensity activities and obtained at least 600 MET-min/week; Inactive: insufficiently active, if they did not meet the requirements for above 2 category.	Female (Odds Ratio (OR): 2.53, Confidence Interval (CI): 1.58 to 4.07), self-employed (OR 2.98, CI 1.78 to 4.99), and blue-collar workers or farmers (OR 2.24, CI 1.27 to 3.95) were more likely to have a higher International Physical Activity Questionnaire (IPAQ) scores and were classified as physically active. High school education didn’t differ for PA compared (OR 1.52, CI 0.94 to 2.46).
Daniels et al., 2018 [43]	Sub-Saharan Africa (Kenya, Zambia, Malawi)	Mixed method	Attitudes, responses, and reactions of HIV-positive women in three sub-Saharan African regions toward a therapeutic exercise intervention, aimed to determine the presence of depression and low body self-image.	*n* = 60 W: 60 (100%) 39 yrs	60 (100%) 60 (100%)	34 (57%)	Qualitative—interview, quantitative—self-report screening questionnaires	Not defined	From a cultural perspective, a percentage of women participants living with HIV (57%) walked (often great distances) and were actively engaged in physical work. Conversely, the majority of participants (53%) reported having a sedentary lifestyle and did not have any previous or current exercise history. Some participants were apprehensive about exercising related to cultural practices and were uncertain about exercise.
Edward et al., 2013 [44]	Nigeria	Cross-sectional	To determine the prevalence of traditional cardiovascular risk factors and the 10-year cardiovascular risk using three risk equations in people living with HIV with no overt vascular disease.	*n* = 265 M: 86 (32%) W: 179 (68%) 38.7 yrs ± 8.7	265 (100%) 214 (80.5%)	Low: 175 (66%)	Questionnaire, medical examination	*PA: Low*—engagement in PA (both recreational and work) that lasted <30 min per day for <3 times per week.	Low physical activity was present in 175 (66%) of the study participants with no significant gender difference. 59 (69%) of males had low physical activity, and 116 (65%) of females had low physical activity.
Frantz and Murenzi, 2013 [45]	Rwanda	Cross-sectional	To determine the anthropometric profile and physical activity levels among people living with HIV and receiving HAART in Kigali, Rwanda.	*n* = 407 M: 94 (23%) W: 313 (77%) 38.82 yrs ± 8.9	407 (100%) 407 (100%)	71 (17%)	Questionnaires, medical profiles, calibrated digital scale and tape	*PA*: According to World Health Organization (WHO) recommendation, a total of at least 30 min of moderate-intensity physical activity per day, five or more days a week. It can also be three or more times per week for at least 20 min of vigorous activity.	Authors reported a significant association between gender and leisure-time physical activity (*p* < 0.05).
Goossens et al., 2020 [46]	Colombia	Mixed method	To elicit patients’ preferences for HIV treatment in the rural population of Colombia.	*n* = 129 M: 80 (64%) W: 39 (31%) 38.4 yrs ± 12.4	129 (100%) NR	NR	Questionnaire	Not defined	Sub-group analysis on education revealed significant differences for all attributes. Conditional relative important attributes for low educated participants were, in descending order, accessibility, physical activity, life expectancy, travel costs, and side effects. However, high educated participants had a different descending order of relative importance, namely: physical activity, life expectancy, side-effects, accessibility, and travel costs.
Gray et al., 2019 [47]	France	Qualitative	To better understand the perceived barriers to and facilitators of PA among French persons living with HIV.	*n* = 15 M: 7 (47%) W: 8 (53%) 46.6 yrs ± 10.3	15 (100%) 15 (100%)	Active: 5 (33%), Seasonal exercisers: 3 (20%)	Semi-structured interviews	*PA*: Physically active, inactive or seasonally active based on WHO recommendations.	Social-environmental barriers to physical activity included: (a) lack of social support, (b) time constraints, (c) financial constraints, (d) climate constraints, and (e) lack of adapted PA offers. Lack of social support included not having someone with whom to exercise. This was mentioned by a less active participants living with HIV as well as active participants. Four socio-environmental facilitators emerged from the analysis that included: (a) social facilitators, (b) social/family responsibilities and activities, (c) adapted PA offers and (d) financial access to PA
Hsieh et al., 2014 [48]	China	Cross-sectional	To examine the associations between osteoporosis-related preventive health behaviors (i.e., physical exercise and dietary intake) and knowledge, self-efficacy and health beliefs in a large cohort of Chinese individuals living with HIV by applying the Health Belief Model (HBM).	*n* = 263 M: 200 (76%) W: 63 (24%) 38.4 yrs ± 9.8	263 (100%) 0 (0%)	Low: 67 (30%); Moderate: 96 (42%), High: 63 (28%)	Questionnaires	Not defined	In the unadjusted analysis, higher levels of physical activity were significantly associated with lower education (OR 0.50, CI 0.27 to 0.91, *p* = 0.024), and higher likelihood of manual labor versus non-manual labor occupation (*p* = 0.002) among participants living with HIV. In the multivariable model, higher levels of PA remained significantly associated with increased likelihood of manual labor versus non-manual labor occupation (Adjusted OR 2.40, CI 1.10 to 5.24, *p* = 0.028). Gender (Adjusted OR 0.85, CI 0.38 to 1.91), smoking (OR 1.49, CI 0.81, 2.77), and alcohol use (OR 1.56, CI 0.75, 3.21) were not significantly related with PA in both models.
Johnson et al., 2015 [49]	United States	Cross-sectional	To investigated an expanded Health Action and Process Approach (HAPA) as a health-promotion model of physical activity/exercise behavior for African Americans living with HIV.	*n* = 110 M: 49 (45%) W: 58 (53%) Transgender: 1 (1%) 46.07 yrs ± 11.02	110 (100%) NR	NR	Questionnaires	Not defined	Social support was not significantly related to PA/exercise (*p* = 0.17).
Johs et al., 2019 [36]	United States	Qualitative	To examine the differences in perceived barriers and benefits of exercise among older people living with HIV by self-identified exercise status.	*n* = 29 M: 25 (86%) W: 4 (14%) M: 57 yrs (median) W: 56 yrs (median)	29 (100%) 29 (100%)	M: 11 exercisers (self-identified) W: 4	Focus groups	*Exercise*: Regular exercise defined as more than 2 days per week, on most weeks.	Authors mentioned the barriers and facilitators of exercise from the perspective of participants living with HIV that included: social support and coping skills, physical environment (gym culture, feeling unsafe in their neighborhood, lack of affordable housing), cost (income), healthy behaviors (drug use).
Kinsey et al., 2009 [50]	South Africa	Cross-sectional	To assess the relationship between CD4 cell count, habitual physical activity levels and functional independence in an HIV-positive South African adult population.	*n* = 186 M: 47(25%) W: 139 (75%) M: 36 yrs ± 7 W: 35 yrs ± 8	186 (100%) 121 (65%)	MET hrs/month: M: *n* = 47; 770 ± 420 W: *n* = 139; 869 ± 443	Questionnaire	Not defined	MET hours/month: males: *n* = 47 770 ± 420, females: *n* = 139 869 ± 443. No significant difference in the total PA levels between the male and female participants living with HIV (*p* = 0.019).
Ley et al., 2015 [51]	South Africa	Intervention study	To explore social-ecological, motivational and volitional correlates of South African women living with HIV with regard to physical activity and participation in a sport and exercise health promotion program.	*n* = 25 W: 25 (100%) Age Range: 20–44 yrs	25 (100%) 13 (52%)	NR	Questionnaires, participatory group discussions, body image pictures, research diaries and individual semi-structured interviews	Not defined	Quotations from the article: “the need for social support or peer support, but essentially trustful and confidential support, from a good friend”; “In the disadvantaged community, for example, women generally are not seen running in the street for exercise and health reasons. This situation might be due to a lack of safety and security in such areas, but also due to social-cultural norms and attitudes about women in the Black African community”; “Transport problems were mentioned, because it was a challenge to come to campus in holidays due to lack of funding (“The school is closed and my aunt is not giving me money for transport and in that way I can’t be present at the gym”).”
Ley and Barrio, 2012 [72]	South Africa	Narrative review	To critically review and discuss opportunities and challenges for benefitting from the different types and effects of physical activity for people living with HIV in the context of South Africa.	NA	NA	NA	Data extraction	NA	Adults living with HIV in South Africa experience various context-specific socio-cultural and economic challenges as well as HIV-related physical, psychological and social constraints affecting engagement in PA.
Mabweazara et al., 2018 [22]	NA	Narrative review	To examine the available literature on physical activity, social support and SES and to generate recommendations for designing and implementing physical activity interventions targeting people living with HIV of low SES.	NA	NA	NA	Data extraction	NA	Results demonstrated that social support plays a major role in promoting PA and counteracting the barriers to PA in people living with HIV of low SES. The results on the role of social support and the influence of SES are integrated to help design appropriate PA interventions for people living with HIV of low SES.
Mabweazara et al., 2019 [52]	South Africa	Cross-sectional	To examine the PA profile of people living with HIV based on PA domains and PA intensity. To determine whether employment status and level of education can predict PA among people living with HIV of low SES.	*n* = 978 M: 218 (78%) W: 760 (22%) 35 yrs ± 8.77	978 (100%) NR	M: Mean (SD) 480.3 min/wk (±583.0) min/wk W: Mean (SD) 269.1 (±331.6) min/wk	Questionnaire, clinical records	Not defined	Men engaged in more PA than women in all domains (work, transport, and leisure) of PA, especially in work-related PA. Overall PA (TMVPA) at 2.5% of the variance (R2, Coefficient of determination = 0.025) tested significant at a 0.01 alpha level (*p* ≤ 0.01). Employment was a significant predictor of overall PA when controlling for age, CD4+ cell count and education level. Education group was not a significant predictor (*p* = 0.06) of overall PA. 2% of the variance (R2 = 0.02) on work-related PA was significant (*p* = 0.01). Employment status was a significant predictor of work-related PA (*p* < 0.01) when controlling for age, CD4+ cell count and level of education. No significant findings were reported for leisure related PA (R2 = 0.01; *p* = 0.25) and transport- related PA (R2 = 0.01; *p* = 0.69).
Mabweazara et al., 2021 [68]	South Africa	Secondary analysis	To determine if age, body weight, height, gender, waist-to-hip ratio (WHR), educational attainment, employment status, CD4+ cell count and body mass index (BMI) can predict overall PA among people living with HIV.	*n* = 978 M: 218 (22%) W: 760 (78%) M: 38.2 yrs ± 8.76 W: 33.9 yrs ± 8.58	978 (100%) 978 (100%)	M: Mean (SD) 480.2 (±582.9) min/wk W: Mean (SD) 369.35 (±222.53) min/wk	Questionnaires	Not defined	Education, employment status and gender significantly predicted total moderate-to-vigorous PA among participants living with HIV. Total moderate-to-vigorous PA was significantly higher in men (mean 480.2 [SD = ±582.9] min/wk) than among women (mean 369.35 [SD ± 222.53] min/wk). Educational attainment (β = 0.127; *p* < 0.01), employment (β = −0.087; *p* = 0.01) and gender (β = 0.235; *p* < 0.01) significantly predicted total moderate-to-vigorous PA. Gender had the greatest association, followed by educational attainment and employment status.
Mangona et al., 2020 [37]	Brazil and Mozambique	Cross-sectional	To compare the PA assessed by accelerometers in women of low SES living with HIV under common antiretroviral therapy (cART) from urban areas of two major cities in South America and sub-Saharan Africa.	*n* = 83 W: 83 (100%) 40.1 yrs ± 6.1 (Brazil) 38.8 yrs ± 8.7 (Mozambique)	83 (100%) 83 (100%)	83 (100%)	Mechanical scale, stadiometer, BMI scale, height scale, tri-axial accelerometer	*PA*: Level of PA: Daily MVPA complied with the American College of Sports Medicine (ACSM) recommendations: sedentary (<30 min/day); moderately active (30–60 min/day); active (>60 min/day). PA intensity: sedentary (0–99 counts/min), light (100–2019 counts/min), moderate (2020–5998 counts/min), vigorous (45,999 counts/min) Daily steps: sedentary (10,000 steps/day); moderately active (10,000-15,000 steps/day); and active (415,000 steps/day)	45% and 22% of women living in Rio de Janeiro and Maputo were sedentary, respectively. PA performed by patients was mostly of light and moderate intensity, while vigorous PA was practically inexistent (3–5 min of the day) and found in only 18% of participants in both cities. Overall, authors reported low levels of PA among women from low SES.
Muronya et al., 2011 [53]	Malawi	Cross-sectional	To obtain data on multiple non-communicable and cardiovascular disease risk factors in adult Malawian adults living with HIV taking ART in an urban setting.	*n* = 174 M: 67 (38%) W: 107 (62%) 40.8 yrs	174 (100%) 174 (100%)	Low PA: 47 (27%)	Questionnaire	*PA*: *Low*—vigorous exercise on fewer than 3 days/wk and doing vigorous or moderate exercise on fewer than 5 days/wk Physical exercise included all activities (both recreational and work) that require physical effort and causes increases in breathing and/or heart rate.	Predetermined criteria for low level of physical activity were fulfilled by 27%. Low PA among men: 25%, Low PA among women: 28%. Adjusted OR for male gender and low PA 0.85, 95% CI (0.4–1.80).
Musumari et al., 2017 [54]	Thailand	Cross-sectional	To describe and document factors related to alcohol use, tobacco smoking, and physical exercise in older adults living with HIV in Chiang Mai, Thailand.	*n* = 364 M: 156 (43%) W: 208 (57%) 57.8 yrs ± 5.6	364 (100%) 362 (98.3%)	215 (59%)	Questionnaires, medical records, onsite clinical examination	*Exercise*: Moderate-intensity activities (activities that require moderate physical effort and cause small increases in breathing or heart rate) or vigorous-intensity activities (activities that require hard physical effort and cause large increases in breathing or heart rate) for at least 10 continuous minutes during free time.	Participants who never attended school were less likely to engage in physical exercises compared to those who had secondary or higher education levels (Adjusted OR, 0.22; 95% CI, 0.08–0.55; *p* = 0.001). Participants with a waist circumference above the normal standards were more likely to report currently engaged in physical exercises (Adjusted OR, 1.96; 95% CI, 1.15–3.34; *p* = 0.01)
Neff et al., 2019 [55]	United States	Qualitative	To examine the barriers and facilitators to exercise among older people living with HIV initiating an exercise regimen.	*n* = 19 M: 19 (100%) 56.9 yrs ± 5.4	19 (100%) 19 (100%)	NR (participants were enrolled in an exercise intervention RCT)	Focus groups	Not defined	Cost was a barrier to initiating and maintaining exercise among participants living with HIV. “Well, first of all, it is the cost. They have many different fee structures which they won’t advertise or let you know of, until they let you walking there and get you into a high pressure sales man.” Motivating factors to initiate exercise was location/availability. “if it’s not going to be convenient, I’m not going to do it.” Social support was an important factor for motivating and continuing exercise.
Nguyen et al., 2017 [56]	United States	Qualitative	To develop an intervention that included both CBT and exercise, first elicited feedback from participants living with HIV to determine what types of exercise therapy would be viewed as feasible and preferred among the HIV community.	*n* = 27 M: 22 (81%) W & transgender: 5 (19%) 54.4 yrs ± 4.8	27 (100%) NR	NR	Focus groups	Not defined	Few participants felt intimidated by the lack of appropriate exercise venues for people living with HIV. One participant stated he did not “feel safe” going to a gym because he believed that he did not fit in at “non-positive” spaces. He felt more comfortable with others who shared his “condition”.
Petróczi et al., 2010 [57]	United Kingdom	Intervention study	To present an analysis of HIV patients with known physical and psychological characteristics to explore associations with non-compliance in prescribed exercise regimes.	*n* = 22 M: 11 (50%) W: 11 (50%) 41.52 yrs ± 7.12	22 (100%) 19 (86%)	22 (100%)	Questionnaire, modified Harvard step test to measure heart rate	Not defined	Adherence to exercise was independent of gender (Chi square = 0.73, *p* = 0.39). In the group of participants who were defined as adherent to exercise, there were 5 female and 7 male patients; whereas among non-adherent group of participants, there were more female (6) than male (4). There was a higher proportion of black African participants living with HIV among the non-adherent group, and higher proportion of white British in the adherent group than expected from the overall ethnic distribution in the sample (Chi square = 9.839, *p* = 0.04).
Quigley et al., 2018 [58]	Canada	Qualitative	To use the Theoretical Domains Framework (TDF) to investigate the barriers and facilitators to participation in exercise of older people living with HIV.	*n* = 12 M: 9 (75%) W: 3 (25%) 56.6 yrs ± 8.8	12 (100%) 12 (100%)	Self-reported PA; High: 7 (58%), Moderate: 3 (25%), Poor: 2 (17%)	Semi-structured interviews	Not defined	Social influence (encouragement from friends and encouragement from health care providers) was a facilitator to PA. Environmental context/resources (cost, weather, lack of facility) was a barrier to PA.
Rehm et al., 2016 [59]	United States	Cross-sectional	To measure PA levels and benefits/barriers to PA in a group of predominantly African-American HIV+ women in the deep south of the United States and determined differences associated with age and depression levels.	*n* = 50 W: 50 (100%) 42 yrs ± 8.8	50 (100%) NR	Vigorous PA: 8 (16%), Moderate PA: 26 (52%) Average steps/day: 7234 (±3075), Active min/wk: 32.5 ±37.7	International Physical Activity Questionnaire Short Form (IPAQ), Fitbit activity monitor, Exercise Benefits and Barriers Scale (EBBS), questionnaire	Steps per day and active minutes per day (activities ≥3 METS): 10,000 steps/day as the cutoff for being considered “active”	Perceived barriers to PA reported by authors included (mean ± SD) of the Exercise Benefits and Barriers Scale (EBBS). Participants were asked to rank their agreement with each of 29 statements (4 = strongly agree, 3 = agree, 2 = disagree, 1 = strongly disagree): “My family members do not encourage me to exercise” [2.34 (±0.9)] ranked as top barriers; “It costs too much to exercise” [1.88 (±0.82)]; “There are too few places for me to exercise” [1.86 (±0.78)].
Roos et al., 2015 [60]	South Africa	Mixed method	To investigate the personal and environmental factors that cause barriers and facilitators of physical activity in a home-based pedometer walking programme as a means of highlighting adherence challenges.	*n* = 42 M: 7 (17%) W: 35 (83%) 38.7 yrs ± 8.9	42 (100%) 42 (100%)	NR	Questionnaires, diary, pedometer	Not defined	Sedentary jobs prevented participants accumulating adequate steps, “The week was challenging in that I was working shifts and I have to sit on a chair the whole shift”. When participants were busy at work, they were less likely to follow their program, “Tired after I was doing the house work and working in the shop the whole weekend”. The state of the weather was frequently voiced as a barrier and complaints ranged from the weather being too hot, cold or raining a lot, “I am fine the weather is the problem”. A social environmental barrier was the incidences of domestic abuse and crime that influenced participants’ lives.An important facilitator to PA was the support and motivation received from friends and family.
Schäfer et al., 2017 [61]	Switzerland	Cohort	To estimate levels of self-reported PA over time by using data from the Swiss HIV Cohort Study (SHCS).	*n* = 8104 M: 5673 (70%) W: 2431 (30%) Median: 45 yrs (IQR: 39.51)	8104 (100%) NR	NR	Questionnaires and clinical	*PA*: *Sedentary*- participants with: (1) free-time PA at most 1–2 times per month and (2) either not working or sedentary activity at work; *Highly active*: participants with: (1) free-time PA at least 5 times per week or (2) intense work-related *PA. Moderately active*: participants not in one of the other two groups.	Authors reported differences in PA between women and men living with HIV. Men living with HIV were more physically active than women living with HIV. Participants with higher completed education reported more free-time PA than those with lower completed education.
Silveira et al., 2018 [62]	Brazil	Cross-sectional	To evaluate the prevalence of physical inactivity and its association with sociodemographic, lifestyle, clinical, anthropometric, and body composition factors in people living with HIV.	*n* = 288 M: 224 (78%) W: 64 (22%) 37.3 yrs ± 11	288 (100%) 198 (69%)	161 (56%)	Questionnaires	*PA: Physical inactivity*: <600 metabolic equivalent minutes/week	Low education (up to 4 years of study) was associated with physical inactivity among participants living with HIV.
Simonik et al., 2016 [21]	Canada	Qualitative	To explore readiness to engage in exercise among people living with HIV and multimorbidity.	*n* = 14 M: 9 (64%) W: 5 (36%) 50 yrs	14 (100%) 14 (100%)	Action and maintenance phases: 4 (29%)	Semi structured interviews	*Exercise readiness* described as a dynamic spectrum ranging from not thinking about exercise, to routinely engaging in daily exercise.	Participants described the importance of social support as facilitating readiness to exercise. Several participants indicated that having someone to exercise with would improve their willingness to engage in exercise. Some described how an HIV-specific exercise program would facilitate their readiness by creating a safe and inclusive environment, eliminating the challenges associated with disclosure. When describing the conditions that influenced readiness to engage in exercise, most participants expressed the importance of accessibility. For some, a perceived lack of financial accessibility created obstacles to engagement and hindered their readiness to exercise.
Vancampfort et al., 2018 [71]	Sub Saharan Africa (South Africa, Ethiopia, Nigeria, Malawi)	Systematic review	To determine the correlates of PA in people living with HIV in sub-Saharan Africa.	*n* = 1015 NR Age Range: 30.5–40.8 yrs	1015 (100%) NR	NR	Chart extraction	NR	Gender differences were inconsistently reported in the included studies, i.e., 2 of 5 studies indicated women living with HIV engaged in more PA than men living with HIV, while 3 other studies showed no difference between genders. No social/cultural factors and physical environment were reported in the included studies.
Vancampfort et al., 2018 [70]	NA	Systematic review	Understanding barriers and facilitators of physical activity participation in persons living with HIV, and reviewing physical activity correlates in people with HIV.	*n* = 13,176 M: 8268 (63%) W: 4908 (37%) Age Range: 30.5–58.2 yrs	13176 (100%) NR	NR	Chart extraction	NR	Higher educational level was associated with higher physical activity levels in 6/7 studies. Gender differences were inconsistently reported, i.e., while six studies indicated men engaged in more physical activity than women, another reported the opposite, while eight other studies showed no difference between genders. While one study reported a higher physical activity levels in the non-white population, another reported lower levels and four studies reported no associations. Having a manual labor versus non-manual labor job and a higher annual income were, all in one study significantly associated with a higher physical activity level while only one of two studies found that having a job was associated with more physical activity. Two of three studies (67%) reported on social support as a potential positive correlate to PA.
Vancampfort et al., 2017 [69]	NA	Systematic review and meta-analysis	To investigate the prevalence and predictors of treatment dropout in PA interventions in people living with HIV.	*n* = 1128 M: 895 (79%) W: 233 (21%) 41.6 yrs	1128 (100%) NR	1128 (100%)	Chart extraction	Not defined	Separate single meta-regression analyses revealed that dropout rates were not moderated by employment status (%), ethnicity/race (% White), smoking status (% smokers). The only exerciser/participant variables that moderated lower dropout rates were a lower percentage of male participants (ß = 1.15, standard error (SE) = 0.49, z = 2.0, *p* = 0.05).
Vancampfort et al., 2019 [38]	Uganda	Cross-sectional	To explore which socio-demographic and clinical variables are associated with compliance with international PA recommendations in people living with HIV in a fishing community in Uganda. Secondary aims were to explore the reasons for and barriers to physical activity.	*n* = 256 M: 77 (30%) W: 177 (70%) 40.5 yrs ± 10.3	256 (100%) 256 (100%)	81 (32%) [According to PA guideline]	Questionnaire, physical examination	*PA*: Moderate to vigorous PA according to recommended target of 150 min/week of moderate to vigorous PA.	Women had a 1.62 (95% confidence interval, CI 1.01 to 2.57) higher odds for not complying with the PA recommendations than men. Those not having a job had a 2.81 (95% CI 2.00 to 3.94) higher odds for not complying with PA recommendations than those having a paid job. Having received any education (yes/no) and presence of AUD (AUDIT ≥ 8) (yes/no) were not statistically significant.
Vancampfort et al., 2020 [63]	Uganda	Cross-sectional	To determine the proportion of adults living with HIV within a Ugandan fishing community in the different PA stages of change according to the trans theoretical model.	*n* = 256 M: 77 (30%) W: 177 (70%) 40.5 yrs ± 10.3	256 (100%) 256 (100%)	NR	Questionnaires	*PA*: Moderate intensity activity for 30 min on most days of the week (e.g., activities that take moderate physical effort and make you breathe somewhat harder than normal). This could include PA during transport, work, household chores, and/or leisure.	No significant differences in employment status, educational status, smoking status, and somatic medication use status were observed between the different stages of change. Reasons for being physically active were to reduce stress among 62 (24%) participants, to reduce feelings of anxiety: 10 (4%) participants, and to reduce alcohol intake: 2 (1%) participant. Barriers to PA included no social support reported by 2 (1%) of participants.
Webel et al., 2015 [64]	United States	Cross-Sectional	To describe patterns of planned exercise implemented in the home setting (i.e., free-living exercise) in people living with HIV by gender and age.	*n* = 102 M: 54 (52.9%) W: 48 (47.1%) 48 yrs ± 8.7	102 (100%) 102 (100%)	NR	Exercise diary, survey, chart extraction	Not defined	Women reported exercising an average of 2.4 h per week, and men exercised an average of 3.5 h per week. No differences in the quantity of exercise between men and women, except during middle adulthood (women = 2.4 h per week, men = 4.5 h per week; *p* = 0.05). When walking was removed, however, this relationship disappeared (Average exercise for women = 1.1 h per wk and men = 4.0 h per wk; *p* = 0.20). Men did less low-intensity walking (Average 4.0 h/wk) than women (average 4.9 h/wk), but this overall difference was not statistically significant (*p* = 0.23). Men and women exercised at different intensities in both young and middle adulthood (*p* = 0.02; *p* = 0.04, respectively). The average exercise frequency for women and men was three bouts per week (*p* = 0.48). No statistically significant differences in the frequency of exercise between men and women participants living with HIV. Removing low-intensity walking significantly decreased the average amount and number of bouts of exercise per week for men and women (all, *p* < 0.01). These findings indicate that women living with HIV may have access to more exercise resources than men or that they are more likely to take advantage of resources, resulting in higher intensity, more balanced exercise patterns.
Webel et al., 2018 [65]	United States	Randomized Control Trial	To evaluate the 3- and 6- month effects of SystemCHANGE on physical activity and dietary quality in people living with HIV at high risk of developing cardiovascular disease.	*n* = 109 M: 69 (64%) W: 36 (32%) Trans: 4 (4%) 53 yrs	107 (100%) 107 (100%)	MVPA [mean (±SD)]: 60.5 (±88) min/wk Sedentary time: 237 (±139) min/day Steps: 6656 (±3191) per day	Survey, chart extraction, daily diary	Not defined	Being female was consistently associated with less physical activity among participants living with HIV.
Webel et al., 2019 [66]	United States, Thailand	Cross-sectional	To describe physical activity and cardiorespiratory fitness by sex and age and examine the association between physical activity and CRF in a diverse sample of people living with HIV.	*n* = 702 M: 397 (57%) W: 274 (39%) O: Transgender male 5 (<1%) Transgender female: 14 (2%) Genderqueer or other 12 (2%) 50.5 yrs ± 11.1	702 (100%) 621 (92%)	NR	Questionnaire, chart extraction, medical testing	Not defined	Men reported engaging in more light and moderate PA compared with women (*p* < 0.05). Participants walked an average of 402 (±104) metres on the 6MWT, with expected differences by sex. However, both men and women achieved similar rates (68% vs 69%, *p* = 0.96) of their sex- and age-predicted distance on the 6MWT. Among women engaging in any vigorous physical activity in the past week, there was a 7.3% increase in achieving their age- and sex- predicted distanced on the 6MWT (*p* < 0.001). After controlling for known covariates, authors did not observe a similar relationship in men.
Wright et al., 2020 [67]	Uganda	Mixed method	To identify the interpersonal, environmental, and sociocultural characteristics that influence physical activity, exercise, and diet.	*n* = 59 M: 19 (32%) W: 40 (68%) 58 yrs ± 7	30 (51%) NR	Persons living with HIV: 20%; HIV negative: 40% (meet WHO recommended guideline)	Semi-structured qualitative interview, Actigraph accelerometer, photovoice	*PA*: Moderate-to vigorous physical activity defined as activity of at least 2690 counts/min for a minimum of 10 consecutive minutes.	Common barriers to exercise were a lack of time, expense, and safety concerns, which were not specific to HIV status. Concerns of safety often manifested as an explanation for limited outdoor exercise. One male participant living with HIV described fear of crime as a primary reason for not exercising outside. This sentiment was echoed by others indicating threats to safety as the reasons for not going outdoors to exercise. Quantitative results suggested a trend toward people living with HIV engaging in less physical activity (*p* = 0.13) compared to people without HIV. Among people living with HIV, men performed higher physical activity than women in the following variables, Median minutes of MVPA in the past week, Median steps per day, and distance on the 6MWT. However, results were not statistically significant.

NA: Not applicable; NR: Not reported; MVPA: Moderate to vigorous physical activity; 6MWT: Six minute walk test; IQR: Interquartile range; SD: Standard deviation; * unless otherwise stated (age range; median age).

**Table 2 ijerph-19-13528-t002:** Social determinants of health (SDOH) identified in each of the 41 included articles examining physical activity or exercise among adults living with HIV.

Study First Author Year of Publication	Social Determinants of Health								
	Culture	Education and Literacy	Employment and Working Conditions	Gender	Healthy Behaviours	Income and Social Status	Physical Environment	Race/Racism	Social Support and Coping Skills
Capili et al., 2014 [39]						X			
Chisati et al., 2020 [34]				X					
Cioe et al., 2019 [35]		X		X				X	
Clingerman, 2004 [40]									X
Cobbing and Chetty, 2019 [41]	X					X			X
Dang et al., 2018 [42]		X	X	X					
Daniels et al., 2018 [43]	X								
Edward et al., 2013 [44]				X					
Frantz and Murenzi, 2013 [45]				X					
Goossens et al., 2020 [46]		X		X					
Gray et al., 2019 [47]						X	X		X
Hsieh et al., 2014 [48]		X	X	X	X				
Johnson et al., 2015 [49]									X
Johs et al., 2019 [36]					X	X	X		X
Kinsey et al., 2009 [50]				X					
Ley et al., 2015 [51]	X					X	X		X
Ley and Barrio, 2012 [72]	X	X				X	X	X	
Mabweazara et al., 2018 [22]						X			X
Mabweazara et al., 2019 [52]		X	X	X					
Mabweazara et al., 2021 [68]		X	X	X					
Mangona et al., 2020 [37]				X					
Muronya et al., 2011 [53]				X					
Musumari et al., 2017 [54]		X		X	X				
Neff et al., 2019 [55]						X	X		X
Nguyen et al., 2017 [56]							X		
Petróczi et al., 2010 [57]				X				X	
Quigley et al., 2018 [58]						X	X		X
Rehm et al., 2016 [59]				X		X	X		X
Roos et al., 2015 [60]			X				X		X
Schäfer et al., 2017 [61]		X		X	X				
Silveira et al., 2018 [62]		X		X	X	X		X	
Simonik et al., 2016 [21]						X	X		X
Vancampfort et al., 2017 [69]			X	X	X			X	
Vancampfort et al., 2018 [71]	X			X			X		
Vancampfort et al., 2018 [70]		X	X	X	X	X	X	X	X
Vancampfort et al., 2019 [38]		X	X		X				X
Vancampfort et al., 2020 [63]		X	X		X				X
Webel et al., 2015 [64]				X					
Webel et al., 2018 [65]				X					
Webel et al., 2019 [66]				X					
Wright et al., 2020 [67]				X		X	X		

Legend: ‘X’ and grey shaded cell indicates SDOH was represented in the article.

## Data Availability

The raw data supporting the conclusions of this article will be made available by the authors, without undue reservation.

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
