# Peer review of "The Role of the Social Determinants of Health on Engagement in Physical Activity or Exercise among Adults Living with HIV: A Scoping Review"

_ijerph, 2022, doi:10.3390/ijerph192013528_

Round 1
Reviewer 1 Report
My comments are included in the attached file.

Reviewer 2 Report
It is a well-written literature synthesis work, which involves a lot of hard work by the authors. The topic is quite interesting, too.
The only visible error I have detected is line 444: “Race / Racism” should be “3.3.9 Race / Racism”. Also, some sentences may be improved.
Author Response
On behalf of our research team, thank you for the detailed review of our manuscript entitled: The Role of the Social Determinants of Health on Engagement in Physical Activity or Exercise among Adults Living with HIV: A Scoping Review for consideration as an original research paper to the International Journal of Environmental Research and Public Health (IJERPH); Special Issue: “Social Determinants of HIV Health and Prevention” with Occupational Safety and Health.
Reviewer #2
It is a well-written literature synthesis work, which involves a lot of hard work by the authors. The topic is quite interesting, too.
The only visible error I have detected is line 444: “Race / Racism” should be “3.3.9 Race / Racism”. Also, some sentences may be improved.Thank you so much for the comment!
- The only visible error I have detected is line 444: “Race / Racism” should be “3.3.9 Race / Racism”. Also, some sentences may be improved.
Thank you. We corrected this error. (Result, Page 10) – Line 447.
Round 2
Reviewer 1 Report
The authors have provided thoughtful and thorough responses to the questions raised by reviewers.